# Spatio-temporal evolution of acoustic emission events and initiation of stress fields in the fracturing of rock mass around a roadway under cyclic high-stress loading

Gang Lei[1], Dawei Wu[2]*, Shengyan Zhu[1]

1 Faculty of Quality Management and Inspection, Yibin University, Yibin, Sichuan, China, 2 Department of Civil Engineering, Chengdu Technological University, Chengdu, Sichuan, China

* wdwmine@163.com

**Data Availability Statement:** All relevant data are within the paper.

**Funding:** This work was financially supported by Yibin College Sailing Project (No.412-2020QH10)

## Abstract

To study fracture mechanisms and initiation of stress fields in the rock mass around a roadway subjected to cyclic stress, a series of loading and unloading tests were conducted on the rock mass around the roadway by using high-precision acoustic emission (AE) monitoring. The results show that intense AE activities occur in a specimen during cyclic load-holding at different levels. With the increase in the number of cycles, the overall stability of the specimen gradually decreases. In the cyclic loading and unloading process, the specimen exhibits a Kaiser effect. As the number of cycles increases, more AE events occur in the unloading stage and a Felicity effect is manifest. The spatial distribution of AE events is related to the stress regime and structure of the specimen, crack propagation in the roadway exhibits directionality due to effects of the principal stress. High stress is conducive to microcrack initiation and propagation in the specimen, which accelerates damage accumulation and macrofracture formation in a rock mass. The research provides a reference for roadway support work and disaster prevention and control in deep mines.

## Introduction

Underground mining destroys the equilibrium of any *in-situ* stress field, resulting in increasing tangential stress in surrounding rock near a goaf, decreasing radial stress, and stress concentration within a certain range into the rock surrounding the goaf. With the continuous advance of the working face, the stress in rock mass around the goaf is constantly redistributed, and the rock surrounding a roadway near a stope can be subjected to complicated loading and unloading stresses [1–3]. Weak planes, such as microcracks, fractures, and joints in surrounding rock can gradually coalesce to form macrocracks; moreover, damage to the rock mass accumulates, thus weakening its mechanical properties. Researchers have investigated the mechanical characteristics and failure mechanisms of rock under different stress paths [4–8]. Lin et al. [9] studied the relationship between rock strength and deformation of salt rock under cyclic load, calculated and analyzed the stress-strain behavior, damage variables and

and the Project supported By Open Fund (PLC 2020043) of State Key Laboratory of Oil and Gas Reservoir Geology and Exploitation (Chengdu University of Technology). The funders provided financial support for the purchase of experimental materials, translation and polishing of the manuscript.

**Competing interests:** The authors have declared that no competing interests exist.

dissipated energy in the deformation and failure process of salt rock, and obtained the failure characteristics and energy changes of triaxial failure of salt rock. Ren et al. [10] found that the residual strain continues to grow with the number of cycles and a polynomial relationship between the peak strength and the initial axial compressive load ratio can be established.Vaneghi [11] conducted cyclic loading and unloading tests on granite and sandstone under uniaxial stress to analyse response characteristics of fatigue failure and fracture mechanisms of rock under different loading amplitudes and stresses. Based on the theory of subcritical crack growth, linear elastic fracture mechanics (LEFM), and Charles equation. Ren et al. [12] carried out a numerical simulation on the relationship between rock strength and deformation of marble under cyclic load, and proposed a new damage constitutive model to describe the mechanical characteristics of rock under cyclic load. Lin et al. [13] carried out a series of fatigue tests for yellow sandstone containing surface cracks. A damage variable was derived based on the strain energy induced by macro cracks. Similarly, this damage variable was substituted into a statistical damage constitutive model.

Under loading along different stress paths, the failure of rock is mainly related to crack generation, propagation, and coalescence therein [14–16]. During microcrack initiation and propagation, that energy stored within can be released in the form of elastic waves, thus generating acoustic emission (AE) events. AE signals include rich information about rock damage and fracture, and the number and energy of AE events reflect the degree of damage to the rock. The spatial location of AE events reveals the location at which microfractures accumulate in rock. In recent years, AE monitoring has been widely used to study the mechanical characteristics and failure mechanisms of rock under stress. Xiong et al. [17] thought early microcrack percolation in compression test is eliminated in fracture toughness test of pervasive-sourced rock fracture and AE topology displays clear inter-event triggering for pervasive-sourced rock fracture, but non-triggering for percolation. A series of rock tests including Brazilian indirect tension test (BITT), three-point bending test (TPBT), modified shear test (MST) and uniaxial compression test (UCT) were conducted to investigate the acoustic emission (AE) characteristics and crack classification during rock fracture by Du et al. [18]. The test results show that the rock fracture process presents an obvious segmented variation feature and has a dramatic increasing period, according to the change trends of AE hits and AE energy characteristic parameters.Through AE monitoring and the moment tensor (MT) method, Liu [19] studied the mechanical behaviour of granite under uniaxial compression, providing an analysis of the spatial-temporal evolution of AE and fracture mechanisms under load. Zhang [20] investigated the spatial distribution of AE events and the damage-evolution process of coal specimens under cyclic load based on the single-link cluster (SLC) method. Furthermore, changes in SLC structure and spatial correlation length under different numbers of load cycles were discussed.

It is necessary to study the AE characteristics of rock in the failure process under different stress paths to reveal failure mechanisms of rock, however, there are few studies on the spatio-temporal evolution of microcracks and stress transfer in the rock mass around a mine roadway subject to high-stress cyclic load. By utilising the high-precision AE monitoring technique, in the present research we conducted a series of damage and fracture tests on the rock surrounding a roadway under high-stress loading and unloading. Furthermore, the spatio-temporal evolution, and clustering, characteristics of microcracks during both loading and unloading were studied. Based on the apparent stress of AE events, the initiation and expansion of stress fields in the rock mass around a roadway during fracturing were analysed and the stability of the surrounding rock was evaluated. The research revealed the internal connection between microcrack evolution and final failure of rock damaged under different loading and unloading stresses.

## Design of test schemes

The failure of a rock after being loaded is essentially a process in which weak surfaces such as microcracks, cracks, and defects gradually aggregate and connect within the rock until they develop into large-scale failure. The mechanism of its formation is similar to the fracture caused by indoor cement mortar test blocks under load, and can be seen as a phenomenon of rock failure. Cement mortar has good uniformity and isotropy, and its strength can be adjusted according to the capacity of the loading equipment. It is often used as a material to simulate engineering rock masses [21, 22]. In this research, cement mortar was taken as a model material to replace rock. According to the size of the loading system and its capacity, the uniaxial compressive strength of the similar material had to be 35 MPa. M425 cement was used and the external dimensions of the model were 300 mm × 300 mm × 250 mm. The circular roadway was located in the centre of the model. To ensure that the design strength of the cement mortar model could meet the expected requirements, it is necessary to test the strength of the same batch of standard cast specimens. The model could not be polished on the surfaces and drilled before reaching the design strength. The circular roadway (with a diameter of 46 mm) was located in the centre of the specimen.

An AE monitoring system (Physical Acoustic Corporation, USA) was used to collect AE signals generated in the fracturing rock mass in real time and record waveforms of the collected events. By converting the collected signals using the built-in A/D convertor and storing them on the hard disk of a computer, the built-in software could be used for follow-up analysis. To capture microfracture signals generated in the specimen during loading, eight Nano30 sensors specific to use in cement grout were adopted in these tests. The response frequency of the sensors ranged from 50 to 400 kHz and each sensor was equipped with a 1220A-AST preamplifier. The sensors were arranged with attention paid to monitoring of the area near the roadway and four sensors were separately arranged in the front and back of the model in a staggered format. The eight AE sensors could cover all areas near the sidewalls of the prefabricated tunnel structure and the layout mode of sensors is shown in Fig 1. Before the test, the sampling frequency, sampling length, and sampling threshold of AE events were set to 10 MHz, 5 k, and 45 dB, respectively. To ensure that the sensors were coupled with the specimen, Vaseline® was smeared on the active face, and then the sensors were fixed to the specimen with rubber bands. A ZLCJS-5000 impact-resistant biaxial loading test system was used for loading. The system could control the transverse stress and longitudinal stress in the loading process and the accuracy and overall stability of load application met all test requirements. The layout of the test model is illustrated in Fig 2. During loading, it was necessary to place shock-reducing modules at the area of contact between the loading ram and the model, to reduce that friction on the end of the model generated during loading.

Four different stress paths were applied: a load-holding path and stress loading and unloading paths under different confining pressures (Fig 3). Under load-holding at different stresses, the confining pressure P2 was separately set to 10 MPa and 20 MPa. P1 and P2 were simultaneously applied at load rate of 3.75 kN/s until the desired stress was reached. When P2 reached 10 MPa (20 MPa), the load was held, while P1 was further increased to the target value of 25 MPa. The load was held until AE signals were no longer generated in the specimen and then the next load-cycle was applied. After three load-cycles, P1 was increased until the specimen was damaged. Under stress loading and unloading, the confining pressure P2 was set to 10 and 20 MPa, and P1 and P2 were simultaneously applied at a rate of 3.75 kN/s until the desired stress was reached. After P2 reached 10 MPa (20 MPa), it was maintained, while P1 was increased to 25 MPa then rapidly decreased. When P1 was equal to P2, the next load cycle was applied. After three load-cycles, P1 was increased to failure. Three intact specimens were tested under each set of stress paths (Table 1).

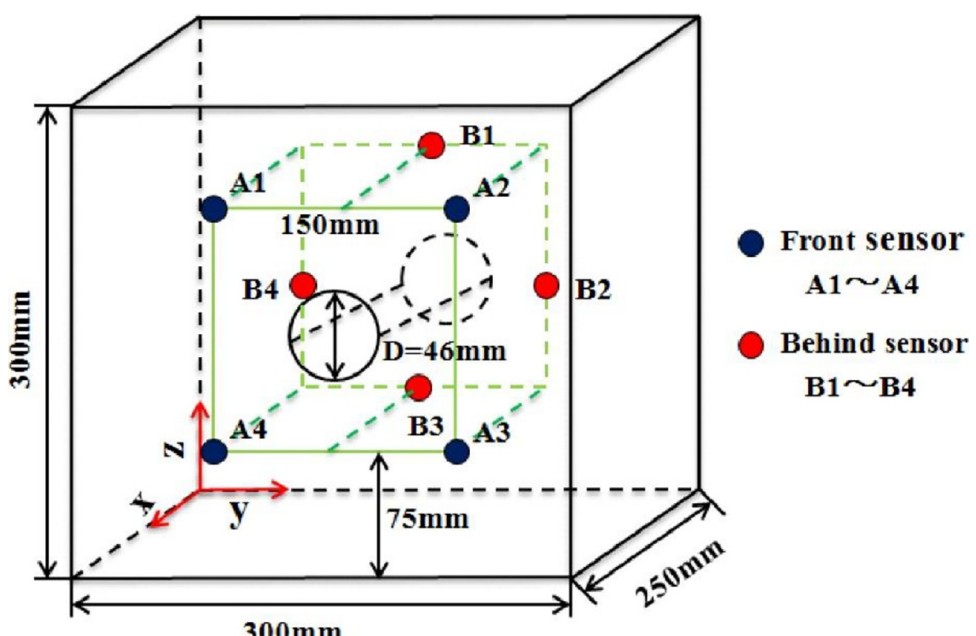

**Fig 1. Arrangement of AE sensors in the specimen.**

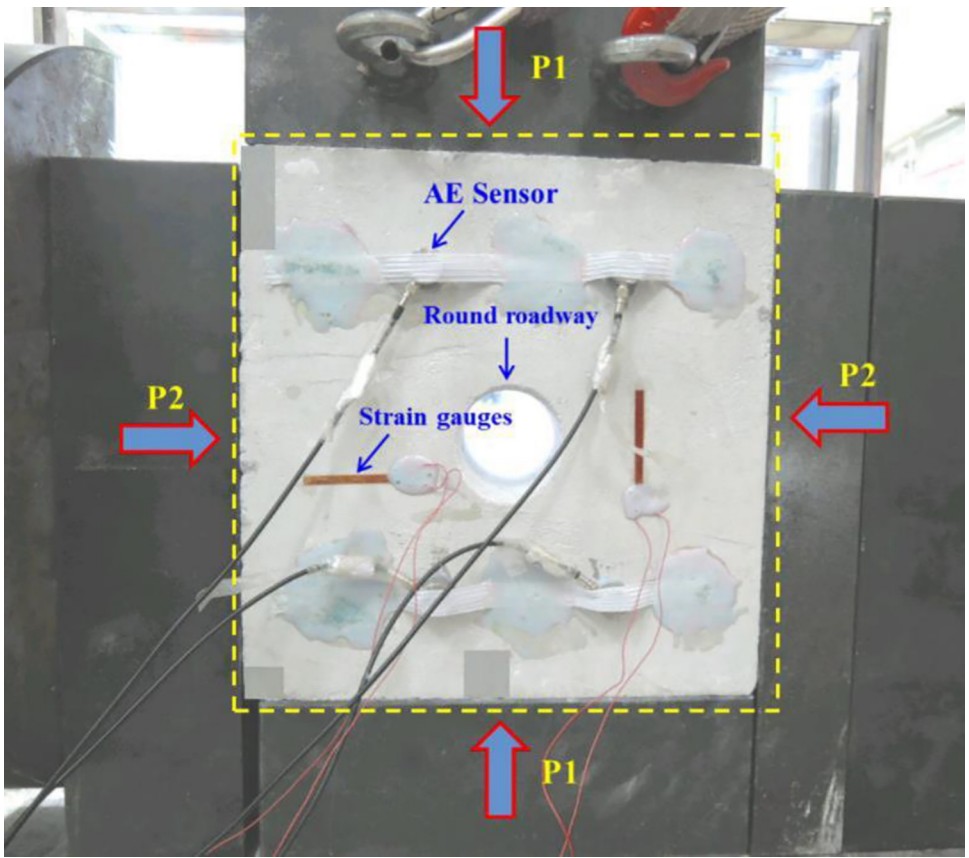

**Fig 2. Test model and loading system.**

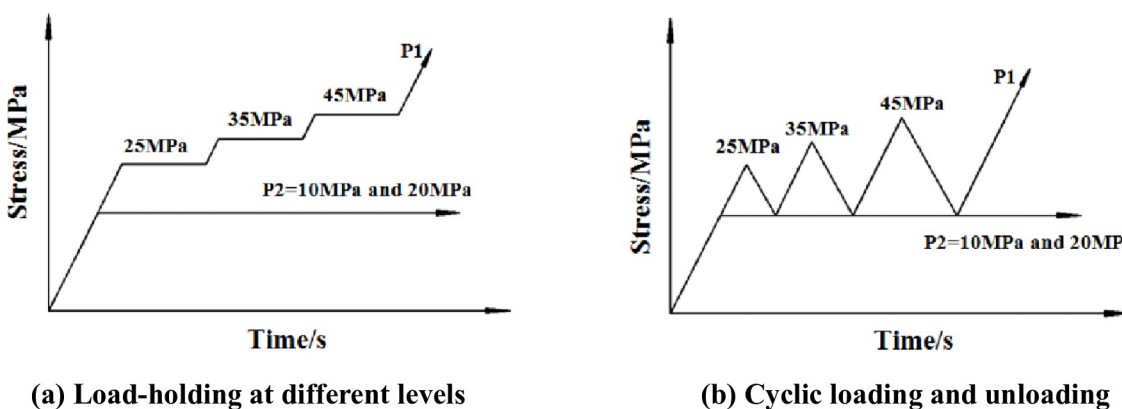

**(a) Load-holding at different levels**     **(b) Cyclic loading and unloading**

**Fig 3. Schematic diagram of loading paths.** (a) Load-holding at different levels. (b) Cyclic loading and unloading.

## AE location method and verification of location accuracy

### Positioning methodology

Positioning of AE sources is based on the travel time difference of P-wave triggered by micro-cracking received by different AE sensors. Common algorithms include least-square method, simultaneous inversion, relative location technique, Geiger's method and simplex method [19, 23–25]. In this paper, Geiger's method is adopted in the calculation of source location. Geiger's method is an iterative approach, with a manually chosen initial value closed to the final result. In each iteration cycle, a correction vector $\Delta\theta$ ($\Delta x$, $\Delta y$, $\Delta z$, $\Delta t$) is calculated based on least-square method. Superposition of the correction vector on the previous iterative point gives the next point. A decision criteria is applied to determine the end of iteration which is stopped when the result satisfy the criteria and the final result is the source coordinates. In each cycle, the source position is calculated by Eq 1.

$$[(x_i - x)^2 + (y_i - y)^2 + (z_i - z)^2]^{\frac{1}{2}} = v_p(t_i - t) \tag{1}$$

where, (x,y,z) is the source coordinate in meter, $(x_i, y_i, z_i)$ is the coordinate of the $i^{th}$ sensor in meter, t is the time of occurrence in second, $t_i$ is the travel time to the $i^{th}$ sensor in second, and $v_p$ is the P-wave velocity in m/s. Geiger's method has been proven to be a reliable approach supported by many AE monitoring practices. Advantages include high accuracy and efficiency, but the method has a strong dependency on the choice of initial value as an inappropriate initial value would lead to divergence. Due to the existence of the prefabricated circular hole, the elastic wave will refract and reflect around the prefabricated circular hole, and the time when the sensor receives the signal will change, which will affect the positioning accuracy. Therefore, when performing acoustic emission localization calculation, the signal that arrives later is deleted, and the five sensors that received the signal first participate in the calculation.

**Table 1. Test schemes.**

| Test plan | Loading type | Model number | P1/ MPa | P2/ MPa | Number of cycles |
|-----------|-------------|--------------|---------|---------|------------------|
| 1 | Load-holding at different levels | S1-1, S1-2, S1-3 | 25, 35, 45 | 10 | 3 |
| 2 | Load-holding at different levels | S2-1, S2-2, S2-3 | 25, 35, 45 | 20 | 3 |
| 3 | Cyclic loading and unloading | S3-1, S3-2, S3-3 | 25, 35, 45 | 10 | 3 |
| 4 | Cyclic loading and unloading | S4-1, S4-2, S4-3 | 25, 35, 45 | 20 | 3 |

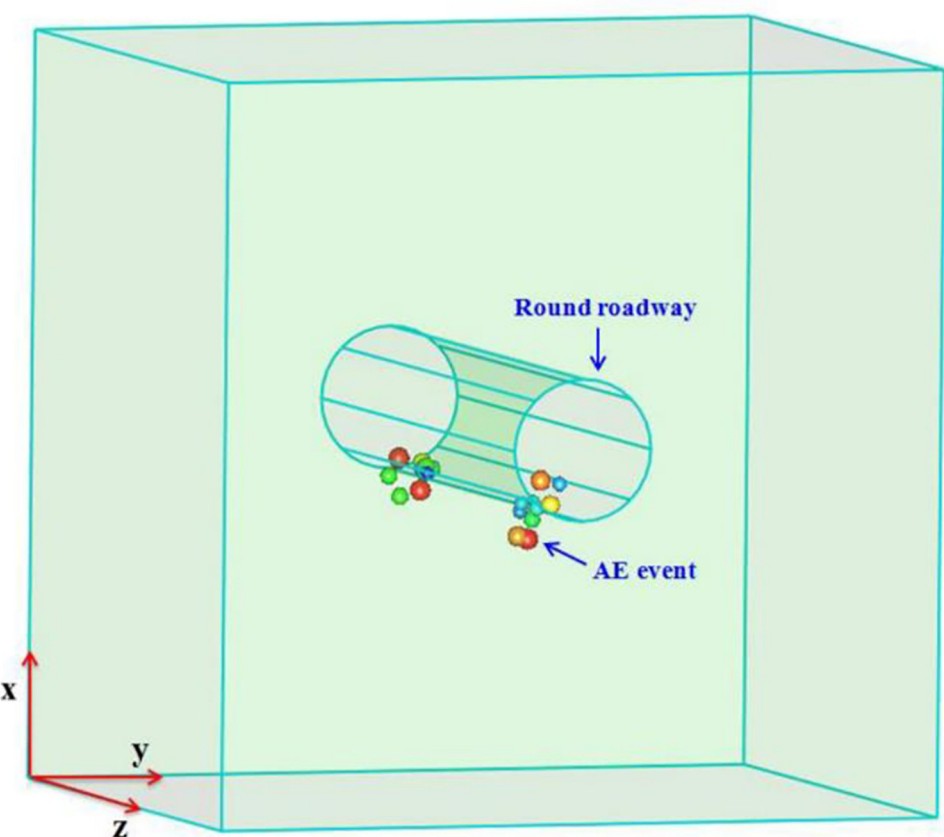

**Fig 4. Manual tapping test positioning result.**

## Precision validation

Precision of source positioning has to be validated after installation of the AE sensors. A common approach is to compare positioning results with actual locations of Manual tapping activities, which provide a useful assessment on the validity of sensor's spatial distribution and performance of the entire system. One of crucial issues influencing the precision is the determination of wave velocity in rock mass, which is affected by cavities, joints, cracks and impurities in rock mass. Hence, multi-directional testing has to be conducted on the rock mass within monitoring range. Regressed P-wave velocity is 4120 m/s. Based on the measured wave velocities, source positioning error can be obtained by comparing the actual coordinates of manual tapping activities and those calculated by monitoring system. A total of 20 manual tapping experiments were conducted. Table 1 lists the statistics of the manual tapping events and positioning results. The maximum deviation is 16.6mm while the minimum is 2.4mm, More than 90% of the positioning error is controlled within 10mm, demonstrating an acceptable coincidence and satisfaction on the monitoring precision. The positioning results are shown in Fig 4.

## Time series features of AE events in rock fracturing

The results of a representative specimen were selected for each test scheme and the specimens were labelled S1-1, S2-3, S3-2, and S4-1. The changes in the AE hit rate and cumulative number of AE hits with time were extracted and the number of AE signals was obtained at a time interval of 1 s. The time series features of AE events are illustrated in Fig 5.

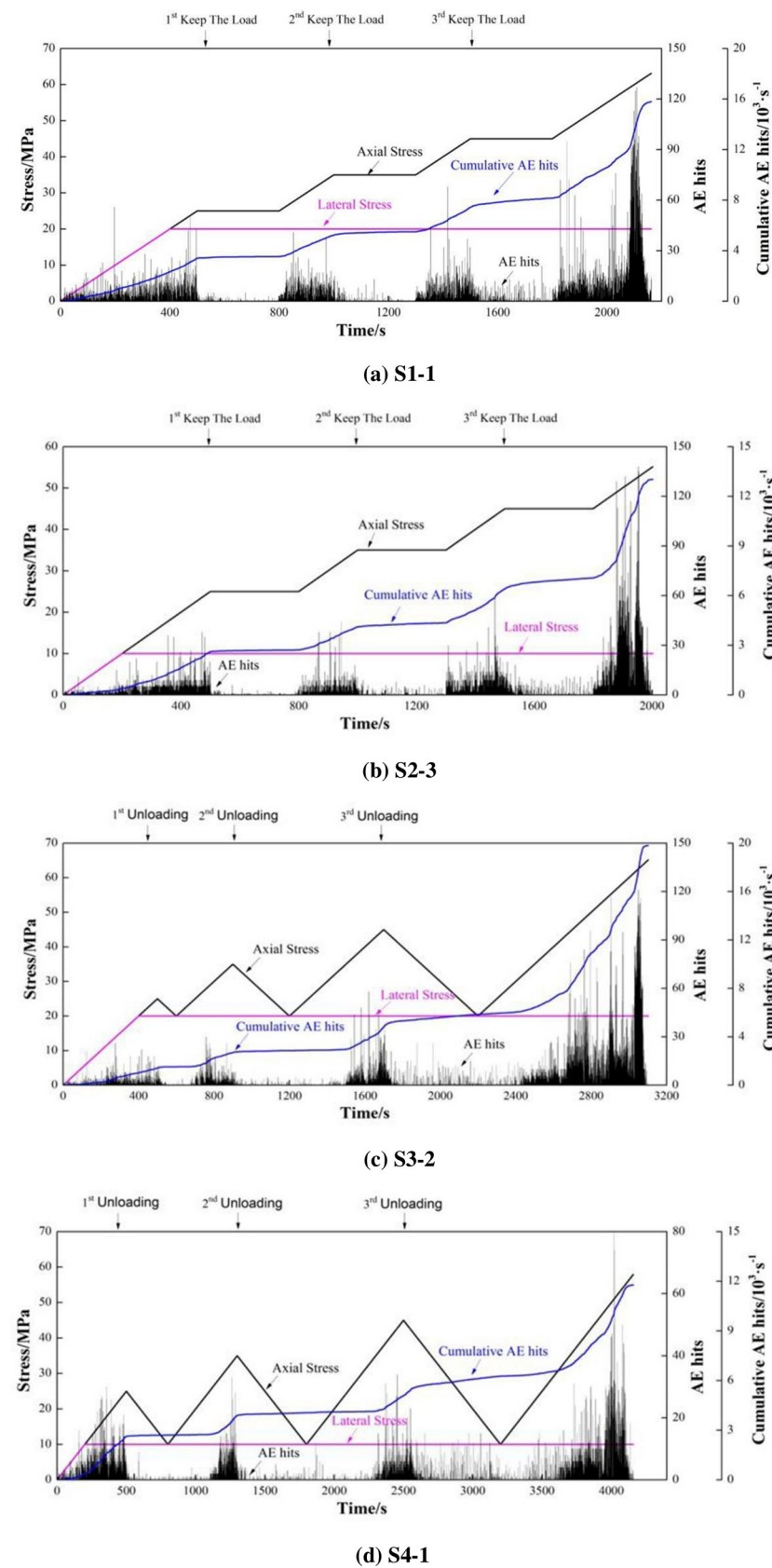

**Fig 5. Time series data: AE events.** (a) S1-1. (b) S2-3. (c) S3-2. (d) S4-1.

As shown in Fig 5(A) and 5(B), the time series of AE events are similar. In the early stages of loading, pores and fractures in the specimen become compacted and close. With increasing stress, the specimen gradually enters a stage of stable crack propagation. After the stress exceeds the crack initiation threshold, the compacted primary cracks begin to propagate, accompanied by the initiation of new cracks, so AE event counts start to rise. As the number of load-cycles rises, the specimen enters its unstable crack propagation stage, in which many new cracks are generated, to then propagate and coalesce. Moreover, the AE activity is further enhanced and cracking of the specimen increases rapidly. With increasing stress, the time taken for AE activities to stabilise increases during load-holding and the overall stability of the specimen is reduced. By comparing results from loading tests on specimens S1-1 and S2-3, different confining pressures (P2) exert significant influences on the strength of the specimen. The lower the confining pressure, the weaker the specimen and the more evident the AE activities in the later stress-loading stage. Fig 5(C) and 5(D) show that the time-series features of AE events are also similar. In the early stress-loading stage, with the increase of stress, cracks in the specimen are gradually closed, while new cracks initiate and propagate (moreover, the AE activity therein is strengthened). When the applied stress reaches the maximum stress in the last cycle, the AE hit rate begins to increase, and the specimen shows an obvious Kaiser effect. After the third cyclic unloading, the level of AE activities remains high and microcracks in the specimen are generated throughout. As a result, the overall stability of the specimen decreases. When P1 is reapplied after the third unloading stage, significant AE events occur before the stress reaches its maximum value in the last cycle and the specimen exhibits a Felicity effect because significant plastic deformation occurs under load and the damage thereto constantly accumulates. After the previous cyclic stress unloading, the specimen does not completely recover from the deformation and a certain residual strain remains. As P1 is reapplied, the load required to reach the previous maximum deformation is smaller than the maximum load in the previous cycle and specimen is weakened. In addition, AE activities have no memory effect. By comparing data from specimens S3-2 and S4-1, the confining pressure inhibits the generation of AE events and decelerates lateral expansion and failure of the specimen: a greater axial stress is needed to reach the peak strain and the time to onset of damage is increased.

Both stress loading and stress holding (or unloading) will cause damage and destruction in the specimen. According to statistics, for S1-1 sample, the proportion of AE signals generated by stress loading and stress holding is 91.32% and 8.68% respectively. For S2-3 sample, the proportion of AE signals generated by stress loading and stress holding is 86.21% and 13.79% respectively. For S3-2 sample, the proportion of AE signals generated by stress loading and stress unloading is 92.25% and 7.75% respectively. For S4-1 specimen, the proportion of AE signals generated by stress loading and stress unloading is 87.52% and 12.48% respectively. The test results of different groups of samples show that the higher the stress level of rock mass is, the higher the AE signal ratio generated by stress holding (or unloading) is, and the stress loading is more likely to damage and destroy the rock mass in high stress state.

## Spatial evolution of AE events during fracturing

AE events produced in the first three load-cycles and the last stage of each test were displayed in three dimensions using Java Debug Interface (JDI) software. The energy of AE events was graded by colour and diameter of circle. The greater the energy, the larger the circle representing the event and the closer the colour to red (Fig 6).

After the first load-cycle, AE events produced in specimen S1-1 are mainly concentrated in the areas around the roadway and in the left and right areas and the lower area thereof. When the lateral pressure P2 is kept constant, with the increase of P1, stress concentration occurs in

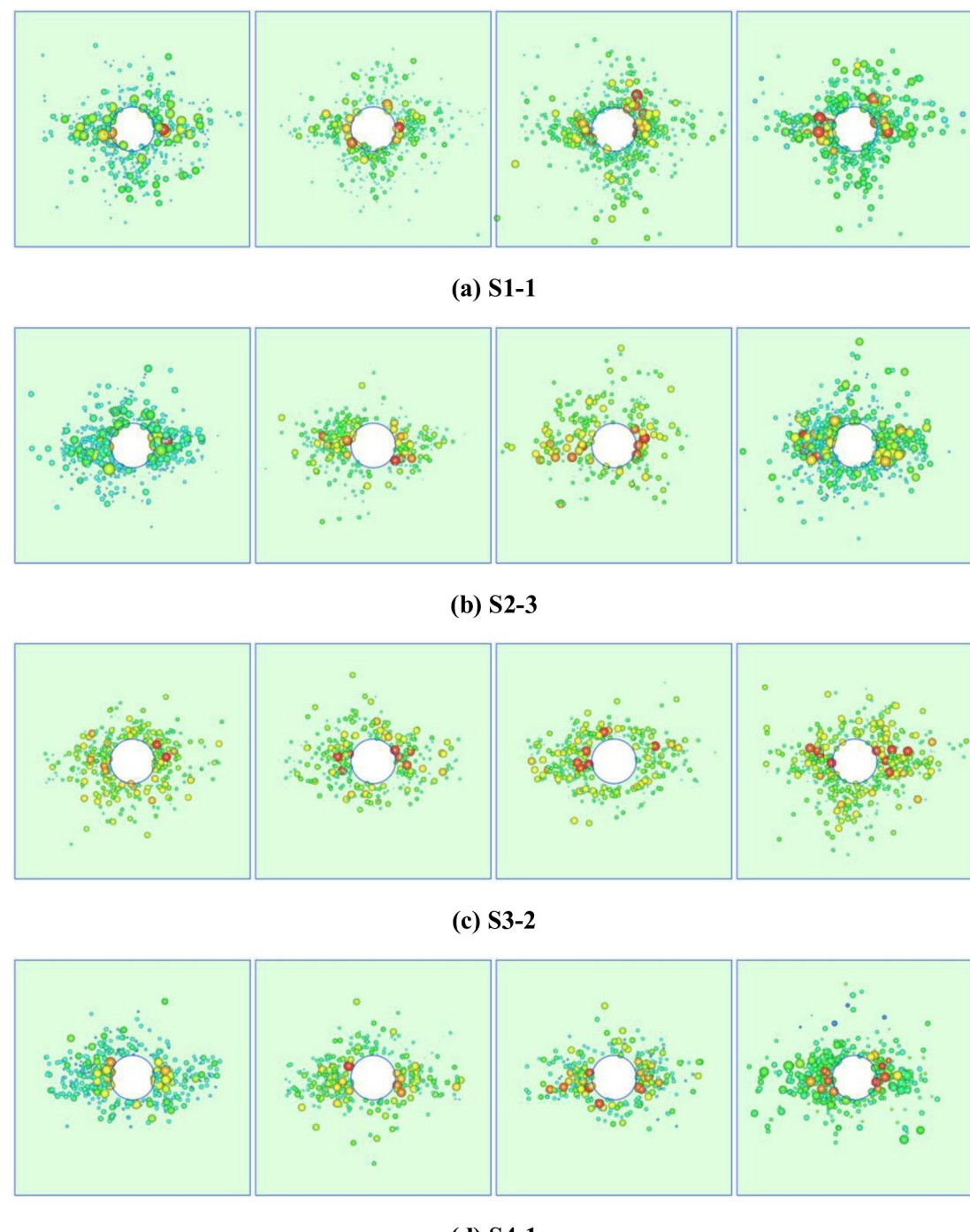

**(a) S1-1**

**(b) S2-3**

**(c) S3-2**

**(d) S4-1**

**Fig 6. Spatial evolution characteristics of cumulative AE events.** (a) S1-1. (b) S2-3. (c) S3-2. (d) S4-1.

the left and right of the roadway; because high stress is conducive to microcrack initiation and propagation in the specimen, AE events are mainly distributed in the left and right areas of the roadway, accompanied by the generation of high-energy events. After the last stress-loading cycle, the number of AE events increases, while the overall stability of the specimen decreases. Moreover, those high-energy events are concentrated in the left and right areas of the roadway.

This indicates that the probability of failure in the left and right areas of the roadway increases. The comparison of the spatial distribution of AE events generated in specimens S2-3 and S1-1 shows that the lower the confining pressure on the specimens, the more concentrated the distribution of AE events in their left and right-hand sides. The spatial distribution of AE events is related to the stress field in, and structure of, each specimen. After the first load-cycle, AE events generated in specimens S3-2 and S4-1 are mainly concentrated in the left and right areas of the roadway and a few high-energy events were recorded. As the number of cycles increases, these high-energy events also occur in the upper and lower areas of the roadway: the reason for this is that, when the specimens are subjected to cyclic application of stress P1, the elastic strain energy constantly accumulates, microcracks are continuously generated and propagate, and damage accumulates. After unloading of P1, stress in the specimens is redistributed, and cracks compacted and closed in the loading stage reopen and propagate in an unsteady manner. The test results after loading of the four specimens demonstrate that AE events are mainly distributed in the left and right areas of the roadway, which is opposite to the direction of the maximum principal stress when the specimens are damaged.

## Stress initiation and transfer in the rock mass around a roadway

Apparent stress, as a dynamic parameter describing the seismic source intensity, conveys abundant information about regional stress states, so is of significance when describing the fracturing process at a seismic source [26–28]. The apparent stress reflects the activity of stress states in an area and represents the level of activity and characteristics of the seismic source. The change in apparent stress can reveal stress accumulation and release in the region around a seismic source. Based on the trend, and spatio-temporal distribution, of the apparent stress, the trend of strong seismic activities and potential danger areas can be analysed and evaluated.

Radiated microseismic energy and seismic moment are the two most important parameters to calculate the apparent stress. The radiated microseismic energy refers to the energy released in the transformation from elastic deformation to inelastic deformation of a rock mass. The seismic moment is a physical quantity characterising the fracture intensity in a rock mass and is the equivalent point source moment of fault and dislocation at the seismic source causing fracture. According to the Brune model [29], the apparent stress can be calculated thus:

$$\sigma_{App} = \mu \frac{E_s}{M_0} \qquad (2)$$

where, $\mu$ represents the shear modulus of media in the seismic source area ($c.$ $3.0 \times 10^4$ MPa); $E$ and $M_0$ denote the wave energy radiated from the seismic source and seismic moment, which can be obtained through analysis and inversion of waveforms. The ratio of the two represents the elastic wave energy radiated per unit seismic moment, so, the apparent stress physically means the product of the elastic wave energy radiated per unit seismic moment and the shear modulus of media in the seismic source area, which is the seismic wave energy radiated per unit dislocation. When inelastic attenuation is neglected, the displacement spectrum of the microseismic source can be expressed by the low-frequency and corner-frequency of the displacement spectrum. The displacement spectrum of the source can be expressed as follows:

$$D(f) = \frac{\Omega_0}{1 + \left(\frac{f}{f_0}\right)^2} \qquad (3)$$

where, $\Omega_0$, $f$, and $f_0$ denote the zero-frequency limit, waveform frequency, and corner frequency of seismic waves, respectively. The velocity spectrum can be calculated according to

the displacement spectrum of the seismic source:

$$V(f) = D(f)2\pi f \tag{4}$$

Based on the Brune model, the integrals of displacement power and velocity power spectra can be calculated thus:

$$S_D = 2\int D(f)^2 df = \frac{1}{4}\Omega_0^2(2\pi f_0) \tag{5}$$

$$S_V = 2\int V(f)^2 df = \frac{1}{4}\Omega_0^2(2\pi f_0)^3 \tag{6}$$

The relationship between the integral of the displacement power spectrum and the integral of the velocity power spectrum is as follows:

$$\frac{S_D}{S_V} = \frac{1}{(2\pi f_0)^2} \tag{7}$$

In accordance with Formulae (3) to (7), the corner frequency and zero-frequency limit of seismic waves can be deduced:

$$f_0 = \frac{1}{2\pi}\sqrt{\frac{S_V}{S_D}} \tag{8}$$

$$\Omega_0 = \sqrt{4S_D^{3/2}S_V^{-1/2}} \tag{9}$$

Therefore, the radiated microseismic energy and seismic moment are given by:

$$E_s = 4\pi\rho\beta S_V \tag{10}$$

$$M_0 = 4\sqrt{\frac{5}{2}}\pi\rho\beta^3\Omega_0 \tag{11}$$

where, $\beta$ and $\rho$ separately represent the wave velocity (km/s) and density (g/cm$^3$) of the rock mass. To eliminate the influences of anomalously high values at several monitoring points (in terms of the apparent stress), logarithmic averaging as proposed by Archuleta *et al*. [30] is used to calculate the average value $x$ of the apparent stress. The value is taken as the calculated result representing a single earthquake.

$$\bar{x} = \exp\left[\frac{1}{N}\sum_{i=1}^{N} In x_i\right] \tag{12}$$

$$x = \left[\frac{1}{N-1}\sum_{i=1}^{N}(In x_i - In\bar{x})^2\right]^{\frac{1}{2}} \tag{13}$$

where, $N$ and $x_i$ denote the number of monitoring points in the calculation and the calculated result for the $i^{th}$ monitoring point, respectively. The apparent stress caused by AE events can be calculated thus: firstly, the parameters and waveform information relating to AE events are extracted, the arrival time of waveforms of rock fracture is corrected by the Akaike information criterion (AIC) algorithm, and the AE location results are calculated by Simplex method.

Secondly, according to the waveform information conveyed by each AE event, the integrals of velocity power and displacement power spectra corresponding to AE events were determined by using a program to calculate the apparent stress. Finally, combined with the location results, the energy released by the corresponding AE event and the seismic moment were calculated, thus giving the apparent stress associated with of AE events generated in the fracturing of the rock mass. By calculating the apparent stress based on AE information from specimens S1-1, S2-3, S3-2, and S4-1, the apparent stress in the first three cycles and the later stage of loading could be obtained (Fig 7).

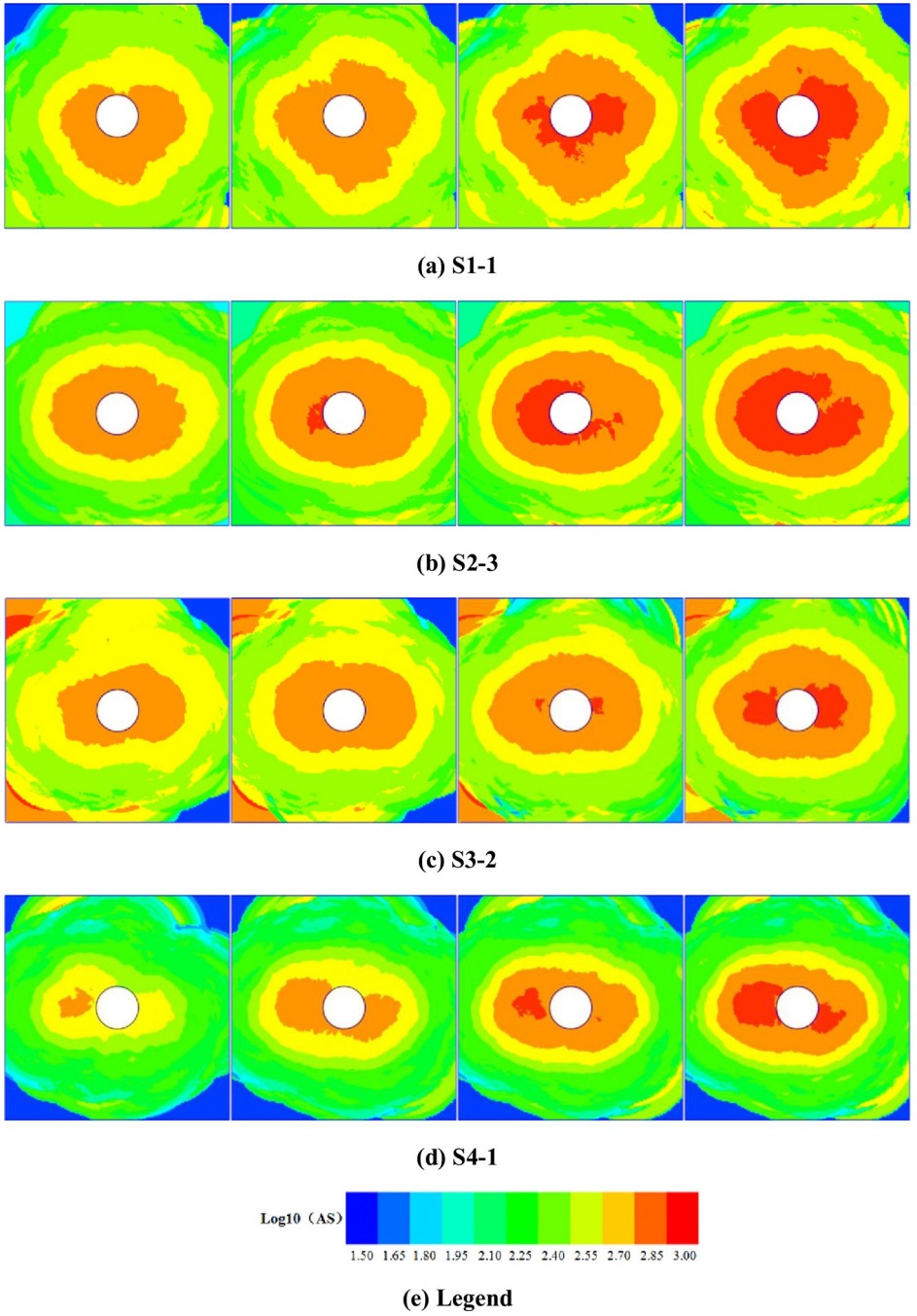

**(a) S1-1**

**(b) S2-3**

**(c) S3-2**

**(d) S4-1**

**(e) Legend**

**Fig 7. Temporal changes in the distribution of apparent stress.** (a) S1-1. (b) S2-3. (c) S3-2. (d) S4-1. (e) Legend.

After the first load cycle, the apparent stress in the surrounding area of the roadway in the specimen S1-1 is large and stress concentration occurs in both sides and floor of the roadway. After the second and third load cycles, the range of influence of the apparent stress concentration around the roadway is increased. After loading, the stress concentration in the surrounding area of the roadway became more obvious and the high apparent stress is mainly distributed around both sides and the floor of the roadway. The range of influence further increases and roadway instability is imminent. After the first load cycle on specimen S2-3, a stress concentration appears around the roadway and forms an oval with its long axis being horizontal, wherein the distribution of apparent stress is highly directional. With the increase of the number of cycles, the stress concentration area in specimen S2-3 expands and significant stress concentration occurs on the left and right-hand sides of the roadway. Compared with the final stress distribution seen in specimen S1-1 at failure, specimen S2-3 is under a lower apparent stress and contains a smaller zone of stress concentration at failure because it is subjected to a lower confining pressure. However, the direction of development of the zone of stress concentration is clearer; the direction in which the expansion of the zone of increased apparent stress in specimen S3-2 is similar to that in specimen S2-3, namely it is opposite to the direction of the maximum principal stress. In the initial loading process, elastic strain energy is constantly stored in the specimen. After the axial stress is reduced, due to lateral stress relief, the specimen expands and deforms in the axial direction, accompanied by the generation of a larger number of AE events. As a result, apparent stress concentration areas are found in the roof and floor of the roadway. Some zones of stress concentration appear in the upper left and lower left corners of specimen S3-2, which may be caused by stress concentration due to an imbalance in the stress-field at the ends of the specimen during loading. The extent of concentration and range of the apparent stress in specimen S4-1 are smaller than those in specimen S3-2, while the zone of stress concentration develops with greater directionality; because different stresses are applied to the specimens, the strain energy stored in different areas of the specimen differs, and the principal stress is applied in the axial direction. Therefore, more elastic strain energy is stored in the left and right-hand areas of the roadway. During the loading and unloading processes, the elastic energy stored in the specimen is gradually released and the distribution of apparent stress becomes increasingly directional.

## Conclusions

By using a high-precision AE monitoring technique, damage and fracture tests on rock surrounding a roadway under different cyclic stress paths were conducted. The spatio-temporal evolution and clustering characteristics of microcracks in the fracturing of such a rock mass were investigated and the initiation and expansion of stress fields during fracturing were analysed. The conclusions are as follows:

1. During cyclic load-holding at different loads, intense AE activities were generated in the specimens after each stress loading cycle and the stress attenuated rapidly after load-holding. With the increase in the number of cycles, the attenuation time increased and the overall stability of the specimen decreased. During cyclic loading and unloading, the specimens in the early stages of loading showed an obvious Kaiser effect. As the number of cycles increased, the number of AE events in the unloading stages increased and an evident Felicity effect was manifest. In the early stages of the loading process, the specimen was in an elastic state of stress, and exhibited obvious memory characteristics. In the later stages of loading, the specimen underwent significant plastic deformation and damage continuously accumulated. After previous cyclic stress unloading, the deformation proved irrecoverable,

leaving a certain residual strain in the specimens. The global strength of the specimens decreased and AE activities generated therein exhibited no memory effect.

2. The spatial distribution of AE events was related to the stress state and structure of the specimens. The high stress was conducive to microcrack initiation and propagation in the specimens. When the longitudinal stress exceeded the transverse stress, AE events were mainly distributed on the left and right-hand sides of the roadway, accompanied by the generation of high-energy events. Crack initiation and propagation in the rock mass around a roadway showed the opposite trend and direction to the maximum principal stress applied to the rock mass and different directions of principal stress had an obvious directional guiding effect on crack propagation in the roadway.

3. When a rock mass was subjected to stress loading and unloading, the internal stress field was constantly adjusted, accompanied by microfracture development and propagation. With the continuous increase of the longitudinal stress, areas of high stress-concentration appeared on the left and right-hand sides of the roadway, where they constantly aggregated and expanded. These zones of high stress were conducive to crack initiation and accelerated the accumulation of damage and the formation of macrofractures in the rock mass. Due to different stresses being applied to the specimens, the strain energy stored in different areas within the specimen differed and more elastic strain energy was stored in the direction opposite to the direction of action of the applied principal stress. In the loading and unloading process, the elastic energy stored in the specimens was gradually released and the distribution of apparent stress became clearly directional.

4. Both stress loading and stress holding (or unloading) will cause damage and destruction in the specimen. According to statistics, for S1-1 sample, the proportion of AE signals generated by stress loading and stress holding is 91.32% and 8.68% respectively. For S2-3 sample, the proportion of AE signals generated by stress loading and stress holding is 86.21% and 13.79% respectively. For S3-2 sample, the proportion of AE signals generated by stress loading and stress unloading is 92.25% and 7.75% respectively. For S4-1 specimen, the proportion of AE signals generated by stress loading and stress unloading is 87.52% and 12.48% respectively. The test results of different groups of samples show that the higher the stress level of rock mass is, the higher the AE signal ratio generated by stress holding (or unloading) is, and the stress loading is more likely to damage and destroy the rock mass in high stress state.

## Author Contributions

**Investigation:** Shengyan Zhu.

**Writing – original draft:** Gang Lei, Shengyan Zhu.

**Writing – review & editing:** Dawei Wu.

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
