## [Decision Letter · Decision Letter 0]

14 Mar 2023

PONE-D-22-35616Spatio-temporal evolution of acoustic emission events and initiation of stress fields in the fracturing of rock mass around a roadway under cyclic high-stress loadingPLOS ONE

Dear Dr. Wu,

Thank you for submitting your manuscript to PLOS ONE. After careful consideration, we feel that it has merit but does not fully meet PLOS ONE’s publication criteria as it currently stands. Therefore, we invite you to submit a revised version of the manuscript that addresses the points raised during the review process.

Please address all the comments. 

We look forward to receiving your revised manuscript.

Kind regards,

Xun Xi

Academic Editor

PLOS ONE

Journal Requirements:

“This work was financially supported by Yibin College Sailing Project (No.412-2020QH10).The Project supported By Open Fund (PLC 2020043) of State Key Laboratory of Oil and Gas Reservoir Geology and Exploitation（Chengdu University of Technology）”

“This work was financially supported by Yibin College Sailing Project (No.412-2020QH10).The Project supported By Open Fund (PLC 2020043) of State Key Laboratory of Oil and Gas Reservoir Geology and Exploitation（Chengdu University of Technology)”

“This work was financially supported by Yibin College Sailing Project (No.412-2020QH10).The Project supported By Open Fund (PLC 2020043) of State Key Laboratory of Oil and Gas Reservoir Geology and Exploitation（Chengdu University of Technology”

Reviewers' comments:

Reviewer's Responses to Questions

**Comments to the Author**

1. Is the manuscript technically sound, and do the data support the conclusions?

Reviewer #1: Yes

Reviewer #2: Yes

2. Has the statistical analysis been performed appropriately and rigorously? 

Reviewer #1: Yes

Reviewer #2: Yes

3. Have the authors made all data underlying the findings in their manuscript fully available?

Reviewer #1: Yes

Reviewer #2: Yes

4. Is the manuscript presented in an intelligible fashion and written in standard English?

Reviewer #1: Yes

Reviewer #2: Yes

5. Review Comments to the Author

Reviewer #1: This paper carried out a novel experiment to investigate the evolution of micro-fractures around a roadway under cyclic loadings. The evolution of micro-fractures and how the stress concentration localization are clearly described. It is worth to publishing this paper.

Details see the attached file.

Reviewer #2: The authors carried out a series of loading and unloading experiments of roadway rock mass with high precision acoustic emission monitoring technology to study the fracture mechanism of roadway rock mass under cyclic load and the breeding process of internal stress field under high stress environment. The study provides reference for roadway support and disaster prevention in deep mine, and is suitable for publication in Plos One journal. Some suggestions are given below:

1. The abstract can be simplified.

2. It is suggested to conduct a novelty search on the references in the paper. Please add relevant research in relevant fields in recent years.

3. It is well known that engineering rock mass has heterogeneous characteristics, and its strength characteristics are also the main factors affecting the spatio-temporal evolution of microcracking or on the initiation, expansion, and transfer characteristics of stress fields in rock. In this paper, M425 cement is used to simulate the engineering rock mass, and the engineering rock mass is regarded as isotropic material with good homogeneity. Please explain whether the test results are representative and whether the conclusions obtained are affected by the cement model.

4. Modify the image format to make the picture clearer.

5. It is necessary to appropriately increase the quantified conclusion.

6. PLOS authors have the option to publish the peer review history of their article (what does this mean?). If published, this will include your full peer review and any attached files.

Reviewer #1: No

Reviewer #2: No

---

## [Author Response · Author response to Decision Letter 0]

12 Apr 2023

See the document Response to Reviewers for details of comments。

---

## [Decision Letter · Decision Letter 1]

7 May 2023

Spatio-temporal evolution of acoustic emission events and initiation of stress fields in the fracturing of rock mass around a roadway under cyclic high-stress loading

PONE-D-22-35616R1

Dear Dr. Wu,

We’re pleased to inform you that your manuscript has been judged scientifically suitable for publication and will be formally accepted for publication once it meets all outstanding technical requirements.

Kind regards,

Xun Xi

Academic Editor

PLOS ONE

Additional Editor Comments (optional):

Reviewers' comments:

Reviewer's Responses to Questions

**Comments to the Author**

1. If the authors have adequately addressed your comments raised in a previous round of review and you feel that this manuscript is now acceptable for publication, you may indicate that here to bypass the “Comments to the Author” section, enter your conflict of interest statement in the “Confidential to Editor” section, and submit your "Accept" recommendation.

Reviewer #1: All comments have been addressed

Reviewer #2: All comments have been addressed

2. Is the manuscript technically sound, and do the data support the conclusions?

Reviewer #1: Yes

Reviewer #2: Yes

3. Has the statistical analysis been performed appropriately and rigorously? 

Reviewer #1: Yes

Reviewer #2: Yes

4. Have the authors made all data underlying the findings in their manuscript fully available?

Reviewer #1: Yes

Reviewer #2: Yes

5. Is the manuscript presented in an intelligible fashion and written in standard English?

Reviewer #1: Yes

Reviewer #2: Yes

6. Review Comments to the Author

Reviewer #1: All the questions have been addressed. Now it is a sound study paper now. I am happy to recommend accept this article.

Reviewer #2: (No Response)

7. PLOS authors have the option to publish the peer review history of their article (what does this mean?). If published, this will include your full peer review and any attached files.

Reviewer #1: No

Reviewer #2: No

---

## [Editor Report · Acceptance letter]

20 Sep 2023

PONE-D-22-35616R1 

Spatio-temporal evolution of acoustic emission events and initiation of stress fields in the fracturing of rock mass around a roadway under cyclic high-stress loading 

Dear Dr. Wu:

I'm pleased to inform you that your manuscript has been deemed suitable for publication in PLOS ONE. Congratulations! Your manuscript is now with our production department. 

Kind regards, 

on behalf of

Dr. Xun Xi 

Academic Editor

PLOS ONE